# Environmental factors of food insecurity in adolescents: A scoping review protocol

**Laura Capitán-Moyano**[1,2]ᵒ*, **Nerea Cañellas-Iniesta**[3]ᵒ, **María Arias-Fernández**[1,2]ᵒ, **Miquel Bennasar-Veny**[1,2,4,5]‡, **Aina M. Yáñez**[1,2,4]‡, **Enrique Castro-Sánchez**[1,6,7]‡

1 Research group on Global Health and Sustainable Human Development, University of the Balearic Islands, Palma de Mallorca, Spain, 2 Department of Nursing and Physiotherapy, University of the Balearic Islands, Palma de Mallorca, Spain, 3 Research group on Studies of Relational Social Work, University of the Balearic Islands, Palma de Mallorca, Spain, 4 Health Research Institute of the Balearic Islands (IdISBa), Palma de Mallorca, Spain, 5 CIBER de Epidemiología y Salud Pública (CIBERESP), Institute of Health Carlos III, Madrid, Spain, 6 Brunel University London, Uxbridge, Middlesex, United Kingdom, 7 NIHR HPRU In Healthcare-Associated Infection and Antimicrobial Resistance, Imperial College London, London, United Kingdom

ᵒ These authors contributed equally to this work.
‡ MBV, AMY and ECS also contributed equally to this work.
* laura.capitan@uib.es

**Data Availability Statement:** No datasets were generated or analysed during the current study. All relevant data from this study will be made available upon study completion.

## Abstract

Food insecurity in recent years has increased worldwide due to many planetary events such as the COVID-19 pandemic, geopolitical conflicts, the climate crisis, and globalization of markets. Adolescents are a particularly vulnerable group to food insecurity, as they enter adulthood with less parental supervision and greater personal autonomy, but less legislative or institutional protection. The experience of food insecurity in adolescents is influenced by several environmental factors at different levels (interpersonal, organizational, community, and societal), although they are not usually addressed in the design of interventions, prioritizing the individual behavioural factors. We present a scoping review protocol for assessing and identifying the environmental factors that could influence adolescents' food insecurity. We used the Preferred Reporting Items for Systematic Reviews and Meta-Analysis Protocols (PRISMA-P) and the PRISMA guidelines for Scoping Reviews (PRISMA-ScR) to prepare the protocol. The search strategy will be performed in the following databases: Pubmed/Medline, EMBASE, Biblioteca Virtual de Salud, EBSCOHost, Scopus, Web of Science, and Cochrane Library Plus. The reference list of the included studies will also be hand-searched. Grey literature will be search through the electronic database Grey Literature Report, and local, provincial, national, and international organisations' websites. Assessment of eligibility after screening of titles, abstract and full text, and the resolution of discrepancies will be performed by three independent reviewers. This scoping review will contribute to refine the "logic model of the problem" which constitutes the first step in the intervention mapping protocol. The "logic model of the problem" from the intervention mapping protocol will serve to classify and analyse the environmental factors. The findings from this review will be presented to relevant stakeholders that have a role in shaping the environmental factors.

**Funding:** The authors received no specific funding for this work.

**Competing interests:** The authors have declared that no competing interests exist.

## Introduction

Food security encompasses the continuous availability of safe and nutritious food that caters to individuals' dietary needs and preferences, while ensuring their physical and economic access for a healthy and active life [1]. This multifaceted concept revolves around four primary dimensions: food availability, economic and physical food access, food utilization, and temporal stability in these dimensions. Additionally, there are ongoing discussions about integrating two supplementary dimensions—agency and sustainability—into this definition [2]. In the context of food security, agency denotes the capacity of individuals and communities to autonomously make choices regarding their food consumption, production, processing, and distribution. It also includes their ability to engage in processes that can influence and transform the food system to better meet their needs [3]. Meanwhile, sustainability entails that the practices within the food system should be geared towards preserving natural resources over the long-term to avoid compromising their enjoyment by current and future generations [3]. The addition of these two dimensions would align food security closer to the concept of food sovereignty, that is the right of people to shape their food policies, following the principles of agency, sustainability, participation and gender equity [4].

Food insecurity, marked by deficiencies in the previously mentioned dimensions [1], worsened globally in 2020, with Europe seeing its first increase since 2014, alongside the longstanding issues in Africa and Asia [2]. Recent factors such as the COVID-19 pandemic, climate crisis, and geopolitical conflicts have amplified food and energy shortages, undermining Sustainable Development Goals (SDGs) [5, 6].

Poverty stands as the primary driver of food insecurity, although they don't always coincide, reflecting their intricate relationship [7]. Gender, ethnicity, class, and immigrant status further complicate this interplay, creating intersecting inequalities [8]. Vulnerable groups like single mothers, ethnically diverse households, and families with children face heightened food insecurity [9].

Another group at great risk are adolescents [10] (people from 10 to 19 years old), as they transition towards adulthood with significant physical, cognitive, emotional and social changes [11], including increased personal autonomy along with reduced parental supervision [12], and less normative or institutional protection. This "second window of opportunity" in development necessitates special attention, particularly in early adolescence (ages 10 to 14), to mitigate the negative consequences of food insecurity [13].

The 2017 UNICEF report found a 12.7% food insecurity prevalence among those under 15 in 2015, varying by country and region. For instance, some Global North countries like Germany, Sweden, or Japan saw food insecurity in one in 20 adolescents, while in Mexico or Turkey, it affected one in three adolescents [14]. Notably, this rate might be underestimated since adolescence spans from 10 to 19 years [11]. Certain adolescent sub-groups, such as those in single-parent families, receiving social welfare, low-income, or low educational backgrounds, are more susceptible to food insecurity [15].

Food insecurity's consequences for children and adolescents encompass physical health issues like wasting, stunting, nutritional deficiencies, anaemia, and asthma [15–17]. Equally important are the less explored impacts on mental and social health [10, 18, 19]. Affected adolescents often struggle academically, leading to school dropout, stigma, and feelings of shame due to food scarcity [10, 20]. In regions with less-established children's rights, families may be compelled to withdraw children from school, push them into early labor, or force premature marriages [20].

These consequences pose significant public health, social, economic, and human rights threats. The costs include direct expenses for food assistance programs and clinical services for

food-related conditions. Indirect costs stem from additional support and special education for children impacted physically or cognitively by food insecurity, affecting their academic performance and readiness for school [21].

Addressing the environmental factors of food insecurity would require attention to their position at the interpersonal, organizational, community and societal level within socio-ecological models [22]. The interpersonal level represents the individuals or groups–family and peers–closely connected to members of the priority population and with potential influence on their health-related behaviour. The organizational level refers to systems with specific objectives and multilevel decision-making processes, for example, schools and the healthcare system. The community level indicates the geographical area that comprises people and organizations; a social place shared by individuals that have a sense of living or working and common elements of values, culture, norms, language, and problems of health and quality of life. Finally, the society represents a larger system that controls several aspects of life and development of their constituent systems [22].

Therefore, it is important to develop and to implement interventions aimed at adolescents, specifically for early adolescents (10–14 years old). The design of these interventions should consider environmental factors and embrace an active role of adolescents.

## Rationale

There is little research on environmental factors that influence food insecurity in adolescents. We conducted a preliminary search on individual and environmental factors, which showcased that behavioural and intrafamilial factors were those most studied. For instance, a recent review [10] summarised behavioural factors of adolescents such as weight-related behaviours, which could be both a cause or a consequence of food insecurity. Other authors have also presented adolescent food insecurity as a risk factor for disordered eating behaviours in adulthood [23], or as coexisting phenomena in adolescence [19]. At the intrafamilial level, the existing literature provides an insight on parenting practices and family dynamics–like using restrictive feeding practices [24], harsh parenting, or not having breakfast altogether [10]–in food insecure households.

Other environmental factors at the organizational, community and societal level seem less frequently addressed. Furthermore, when environmental factors are tackled, it is usually from a broader perspective or on very specific populations, without analysing their impact on young adults and adolescents. There is currently no published synthesis on the role and influence of environmental factors on food insecurity among adolescents from a global lens.

The product of the scoping review will be part of a holistic analysis that will include the other factors mentioned above. Later, this analysis will be used to develop a multi-level intervention to address adolescent food insecurity.

## Methods and analysis

### Protocol design

The scoping review will employ the approach described by Arksey and O'Malley consisting of five stages: 1) formulating the research questions, 2) identifying relevant studies, 3) selecting eligible studies, 4) charting the data and, 5) collating, summarising, and reporting the results [25]. The protocol was registered prospectively with the Open Science Framework on 31 October 2022 on the website (https://osf.io/gjqpe). We used the Preferred Reporting Items for Systematic Reviews and Meta-Analysis Protocols (PRISMA-P) [26], and the PRISMA guidelines for Scoping Reviews (PRISMA-ScR) [27] to prepare the protocol (S1 Fig).

This review is the first step to understand the phenomenon of adolescent food insecurity from a holistic perspective. It will take into account the environmental factors surrounding adolescents, and not only individual aspects.

## Conceptual model: "Logic model of the problem"

This scoping review will use the "logic model of the problem" from the Intervention Mapping (IM) protocol to identify and assess the environmental factors that influence adolescents' food insecurity.

IM is a systematic and detailed methodology for the development, implementation, and evaluation of health promotion interventions step by step. This methodology is based on community participation research methods and a multilevel ecological approach, and aims to comprehend and intervene in health and social problems [28]. The IM process includes six steps (logic model of the problem, logic model of change, program design, program production, program implementation plan and, evaluation plan). The first step consists in the analysis of health, quality of life, behaviours and environmental factors that affect the health problem directly or its risk behaviours. Specifically, the environmental factors are divided in four groups: interpersonal, organizational, community, and societal [28]. Fig 1 depicts how the scoping review will inform the environmental factors from the "logic model of the problem".

## Objectives

The main objective is the understanding of the environmental factors that influence the experience of food insecurity in adolescents, following the structure of the "logic model of the problem" proposed in the IM methodology. Other secondary objectives include:

- To identify the environmental factors (interpersonal, organizational, community and societal) that affect food insecurity in adolescents.

- To analyze the mediating role of these factors on the food insecurity experienced by adolescents.

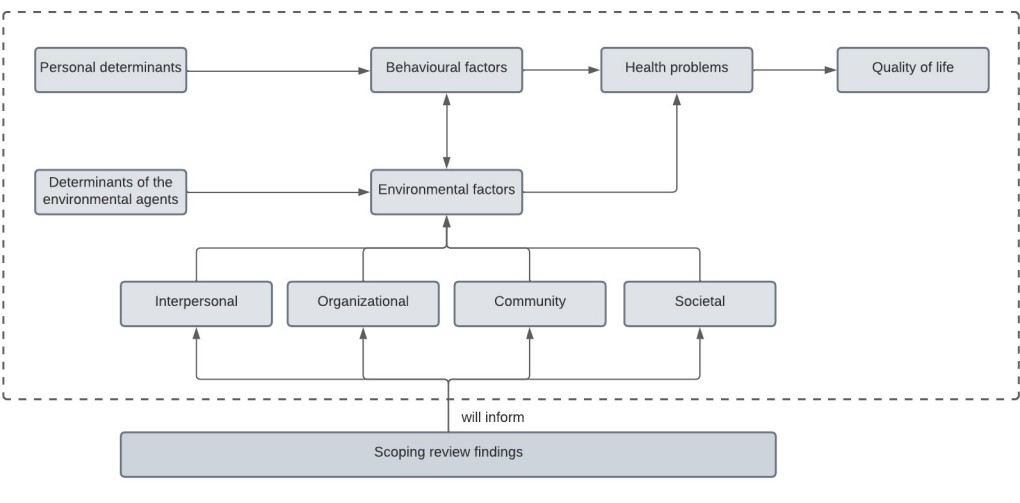

**Fig 1. Logic model of the problem and the contribution of the scoping review on the logic model.** Adapted from the intervention mapping protocol [22].

- To describe the interventions addressing environmental factors of food insecurity, their gaps, limitations, and opportunities.

### Search strategy

To identify relevant studies, a search will be performed in the following electronic databases: Pubmed/Medline, EMBASE, Biblioteca Virtual de Salud (BVS), EBSCOHost, Scopus, Web of Science (WoS), Cochrane Library Plus. The reference list of studies included in the review will also be hand-searched to identify further relevant papers. To avoid a narrow biomedical focus on the review, we will include WoS or Scopus databases as they incorporate a wide range of disciplines.

To ensure that all relevant information is gathered, grey literature will also be searched via Grey Literature Report, to identify pertinent studies, reports, and conference abstracts. We will also conduct a targeted search of the grey literature on websites of local, provincial, national, and international organisations as well as specific organizations which could include information about food systems, food industry, or food sovereignty.

To perform the search strategy, we will combine terms from three domains: food insecurity as the main theoretical concept (e.g., food insecurity, food security, food access, food supply, food poverty), adolescents as target population (e.g., adolescent, children), and environmental factors (intrafamilial, organizational, community, and societal). These terms will be searched as keywords in the title and/or abstract, and subject headings as appropriate. The detailed search strategy is presented in S1 Table.

In our view, food (in)security is wider than food supply, since food availability it is one of the six dimensions that comprise food security. In the MESH thesaurus, however, the hierarchy of these terms is inverted. Also, regarding the target population we decided to add the subject heading "child", to ensure inclusion of the early adolescence period from ten to thirteen years of age.

### Study selection

The search strategy will be limited to literature published in English, Portuguese, or Spanish as our research team is fluent only in these languages. No publication date limit will be applied. The review will include published studies (without restriction on study design) that provide information about the environmental factors of food insecurity among adolescents. The exclusion criteria include published literature that tackle environmental factors on adults or on very specific populations (for example, refugees, people with disabilities, people with infectious diseases). Studies reporting only on consequences of food insecurity, or that only provide information about individual behavioural factors or food safety, will also be excluded.

Once the search phase is over, the results will be exported to the literature management software Endnote 20 (Clarivate analytics, Philadelphia, USA) to deduplicate results and refine references. Subsequently, the data will be exported to the Covidence platform (https://www.covidence.org/) for management and data extraction. Three independent reviewers will review titles and abstracts of studies for initial selection. Later, selected studies will go through a review of full text and data extraction by the same reviewers, that will also meet regularly to discuss and resolve any disagreements. The results of the search will be detailed in the final report and presented in a flowchart ("PRISMA").

### Data extraction and quality appraisal

Data extraction will be performed using a standardized data collection form. The following information will be collected: first author, publication year, country or region, type of

publication, study design, main aim, definition used for food security or food insecurity, participants, sample size, and study findings. The form will also collect data on the environmental factors (interpersonal, organizational, community and, societal): specific groups or general population, and if the context of the study is the Global South, Global North, or both. The Global North includes societies from Europe, North America, Australia, Israel, South Africa, among others, that are developed in terms of wealth and technology, politically stable, and almost null population growth. Whereas the Global South represents societies with a history of colonialism, neo-imperialism, and that maintain large inequalities in decent standards of living, life expectancy, and access to resources [29]. The quality of studies included in the review will be assessed using the Critical Appraisal Skills Programme (CASP) [30].

## Data analysis

At this point, the scoping review will provide an aggregated synthesis of the evidence using the four types of environmental factors described in the logic model of the problem of the IM protocol. Social determinants such as gender and ethnicity will be assessed out of the logic model of the problem, because they are considered as a non-modifiable factor. The scoping review will also present our experiences and recommendations to inform policies and healthcare professionals, and this information will be used to complete the logic model of the problem which is a crucial step in the IM protocol.

## Discussion

This study will constitute the first step of a multi-step research programme aimed at developing a community-based intervention to reduce food insecurity in adolescents. This information will also enable healthcare organisations and policy makers to appraise the environmental factors that play a role in the food insecurity of adolescents. These factors will be classified following the "logic model of the problem" proposed in the IM methodology, analysing whether they are tackled from a broad perspective, or specifically on the adolescent population. Later, the evidence gathered from this review synthesis will be complemented with the experiences and knowledge from relevant stakeholders.

## Limitations

Our review uses a scope on the environmental factors that are situated on a macro level, but we could find that the literature is just focused on individual factors, limiting the pool of papers available for the review. Similarly, there may be few papers focused on the relationship between environmental factors and food (in)security among adolescents. Finally, the findings of the review may be affected by publication bias, thus the choice of a scoping review.

## Supporting information

**S1 Fig. PRISMA-ScR checklist.**
(DOCX)

**S1 Table. Detailed search strategy.**
(DOCX)

## Author Contributions

**Conceptualization:** Laura Capitán-Moyano.

**Investigation:** Laura Capitán-Moyano, Nerea Cañellas-Iniesta.

**Methodology:** Laura Capitán-Moyano, Nerea Cañellas-Iniesta, María Arias-Fernández.

**Project administration:** Laura Capitán-Moyano.

**Supervision:** Miquel Bennasar-Veny, Aina M. Yáñez, Enrique Castro-Sánchez.

**Writing – original draft:** Laura Capitán-Moyano.

**Writing – review & editing:** Laura Capitán-Moyano, Nerea Cañellas-Iniesta, María Arias-Fernández, Miquel Bennasar-Veny, Aina M. Yáñez, Enrique Castro-Sánchez.

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
