## [Decision Letter · Decision Letter 0]

18 Aug 2023

PONE-D-22-33353Environmental factors of food insecurity in adolescents: a scoping review protocolPLOS ONE

Dear Dr. Capitán Moyano,

Thank you for submitting your manuscript to PLOS ONE. After careful consideration, we feel that it has merit but does not fully meet PLOS ONE’s publication criteria as it currently stands. Therefore, we invite you to submit a revised version of the manuscript that addresses the points raised during the review process.

We look forward to receiving your revised manuscript.

Kind regards,

Sanjit Sarkar

Academic Editor

PLOS ONE

Journal Requirements:

Reviewers' comments:

Reviewer's Responses to Questions

**Comments to the Author**

1. Does the manuscript provide a valid rationale for the proposed study, with clearly identified and justified research questions?

Reviewer #1: Partly

Reviewer #2: Partly

2. Is the protocol technically sound and planned in a manner that will lead to a meaningful outcome and allow testing the stated hypotheses?

Reviewer #1: Partly

Reviewer #2: Partly

3. Is the methodology feasible and described in sufficient detail to allow the work to be replicable?

Reviewer #1: Yes

Reviewer #2: No

4. Have the authors described where all data underlying the findings will be made available when the study is complete?

Reviewer #1: Yes

Reviewer #2: No

5. Is the manuscript presented in an intelligible fashion and written in standard English?

Reviewer #1: No

Reviewer #2: Yes

6. Review Comments to the Author

You may also provide optional suggestions and comments to authors that they might find helpful in planning their study.

Reviewer #1: Recommendations

L 28 such as the pandemic, geopolitical conflicts, the climate crisis, and globalization of

Recommend: the pandemic should be clarified. If authors are referring to COVID-19 they should state so. If pandemics in general it should show plural pandemics

markets. Adolescents are a particularly vulnerable group to food insecurity, as they enter29

adulthood with less parental supervision and greater personal autonomy, but les

Recommend: Adolescents gets defined.

L31-34 The experience of food insecurity in adolescents is31

influenced by several environmental factors at different levels (interpersonal,32

organizational, community, and societal), although they are not usually addressed in the33

design of interventions, prioritizing the individual behavioural factors.

Recommend: A reference/s are provided for this sentence.

L 39-40: We used the Preferred Reporting Items for Systematic Reviews and Meta-Analysis39 Protocols and the PRISMA guidelines for Scoping Reviews to prepare the protocol.

Recommend: The Preferred Reporting Items for Systematic Reviews and Meta-Analysis (PRISMA). Insert acronym.

The Preferred Reporting Items for Systematic Reviews and Meta-Analysis Protocols (PRISMA-P) and the PRISMA guidelines for Scoping Reviews (PRISMA-ScR) directed the development of the scoping review protocol.

This is shown much later

L 137: Preferred Reporting Items for Systematic Reviews and Meta-Analysis Protocols137

(PRISMA-P) [24], and the PRISMA guidelines for Scoping Reviews (PRISMA-ScR)

L46-47 will be assessed for eligibility by three46

independent reviewers and discrepancies will be resolved by consensus.

Seeking clarity: will consensus be from all 3 independent reviewers?

L80: Sustainable Development Goals

Recommend: Inserting an acronym as well. Sustainable Development Goals (SDGs)

A definition of adolescents needs to be inserted into the introduction.

L 127-8 There is currently no published synthesis127

on the effect of environmental factors on food insecurity among adolescents

Seeking clarity: Is this so globally? In Europe?

The objectives and the research questions should be separated.

Unsure if all study designs will be included.

L 216-27: The search strategy will be limited to literature published in English, Portuguese, or216 Spanish

Recommend: Insert a reason for only these three (3) language being included.

L229 - : reviewers will review titles and abstracts of studies for initial selection. Later, selected229

studies will go through a review of full text and data extraction by the same reviewers.230

The evaluators will regularly meet to discuss and resolve any disagreements. The results231

of the search will be detailed in the final report and presented in a flowchart (“PRISMA”).

Seeking clarity: Will all 3 reviewers resolve conflict/disagreements?

Reviewers, evaluators??? Try to be consistent

This article should be sent to an editor/proofreader prior to it being resubmitted.

L151 - The first step consists in the analysis

Additional information regarding screeners/funding etc could be inserted

Reviewer #2: Review Report of the scoping review protocol titled ‘Environmental factors of food insecurity in adolescents: a scoping review protocol.’

The authors have effectively positioned their subject (topic) as a relevant academic and policy endeavour by appropriately emphasizing its significance in the introduction and rationale of the study protocol.

They have substantiated the significance of comprehending environmental factors contributing to food insecurity in adolescents by noting that adolescence is a period of transition to adulthood characterized by reduced parental supervision and increased personal autonomy yet lacks comprehensive legislative or institutional safeguards.

Additionally, the authors have highlighted potential repercussions of food insecurity among adolescents. These consequences encompass physical health challenges such as wasting, stunting, nutritional deficiencies, anemia, and asthma, as well as impacts on academic performance and the risk of school dropout. Moreover, the protocol underscores the vulnerability of food-insecure adolescents to being compelled into child labor and early marriage, etc.

The reviewer recommends providing a more comprehensive explanation regarding the significance of this period, along with providing a clear insight into the potential ramifications of 'food stress' during this phase. This should be examined from both a public health perspective and an economic angle. The authors could elaborate on their argument along the following lines.

While early childhood is often recognized as the primary "window of opportunity" for learning and growth, adolescence introduces a distinct phase (referred to as the "second window of opportunity") that presents another critical timeframe for interventions. Throughout adolescence, individuals undergo rapid physical, cognitive, emotional, and social transformations. Consequently, any stress, including challenges related to food availability (food insecurity), and unhealthy eating habits experienced during this life stage can yield numerous health consequences (such as non-communicable diseases, overweight/obesity, etc.).

The Method section appears to lack a well-structured organization in terms of its flow, structure, and content. The authors could have enhanced clarity by distinctly outlining each section to create a more formal protocol-like structure. Instead of writing as stage 1, stage 2, etc. the following sections could have been incorporated:

1. Objectives and Questions

2. Protocol Registration

3. Search Strategy

4. Inclusion and Exclusion Criteria

5. Quality Appraisal

6. Data Extraction

7. Data Analysis

7. PLOS authors have the option to publish the peer review history of their article (what does this mean?). If published, this will include your full peer review and any attached files.

Reviewer #1: **Yes: **Delarise M Mulqueeny

Reviewer #2: No

---

## [Author Response · Author response to Decision Letter 0]

29 Sep 2023

Editorial comments:

R: Based on the PLOS ONE’s style requirements, we removed the bold from the title and added superscripts indicating the contributions by authorship.

2. Please review your reference list to ensure that it is complete and correct. If you have cited papers that have been retracted, please include the rationale for doing so in the manuscript text or remove these references and replace them with relevant current references. Any changes to the reference list should be mentioned in the rebuttal letter that accompanies your revised manuscript. If you need to cite a retracted article, indicate the article’s retracted status in the References list and also include a citation and full reference for the retraction notice.

R: Based on the PLOS ONE’s recommendation, we revised the full reference list and fixed some mistakes of the reference style.

Reviewers' comments:

Reviewer #1: 

Comment #1:

L 28 such as the pandemic, geopolitical conflicts, the climate crisis, and globalization of…

Recommend: the pandemic should be clarified. If authors are referring to COVID-19 they should state so. If pandemics in general, it should show plural pandemics

R: We thank the reviewer for pointing this out. We have clarified that we are referring to COVID-19 pandemic. Therefore, we have made clear, now it reads like this in L27 and L70:

Food insecurity in recent years has increased worldwide due to many planetary events such as the COVID-19 pandemic, geopolitical conflicts, the climate crisis, and globalization of markets.

Recent factors such as the COVID-19 pandemic, climate crisis, and geopolitical conflicts have amplified food and energy shortages, undermining Sustainable Development Goals (SDGs) 

Comment #2:

markets. Adolescents are a particularly vulnerable group to food insecurity, as they enter29 adulthood with less parental supervision and greater personal autonomy, but les

Recommend: Adolescents gets defined.

R: We have introduced a clear definition of adolescent in the introduction. Now it reads like this in L80:

Another group at great risk are adolescents [11] (people from 10 to 19 years old), as they transition towards adulthood with significant physical, cognitive, emotional and social changes [12], including increased personal autonomy along with reduced parental supervision [13], and less normative or institutional protection. This "second window of opportunity" in development necessitates special attention, particularly in early adolescence (ages 10 to 14), to mitigate the negative consequences of food insecurity [14].

Comment #3: 

L31-34 The experience of food insecurity in adolescents is31 influenced by several environmental factors at different levels (interpersonal,32 organizational, community, and societal), although they are not usually addressed in the33 design of interventions, prioritizing the individual behavioral factors. 

Recommend: A reference/s are provided for this sentence.

R: We thank the reviewer for these suggestions. However, the sentence referred to is in the abstract. Therefore, we cannot add a reference, but this is explained further in the introduction with the respective references.

Comment #4:

L 39-40: We used the Preferred Reporting Items for Systematic Reviews and Meta-Analysis39 Protocols and the PRISMA guidelines for Scoping Reviews to prepare the protocol.

Recommend: The Preferred Reporting Items for Systematic Reviews and Meta-Analysis (PRISMA). Insert acronym.

The Preferred Reporting Items for Systematic Reviews and Meta-Analysis Protocols (PRISMA-P) and the PRISMA guidelines for Scoping Reviews (PRISMA-ScR) directed the development of the scoping review protocol.

This is shown much later

L 137: Preferred Reporting Items for Systematic Reviews and Meta-Analysis Protocols137 (PRISMA-P) [24], and the PRISMA guidelines for Scoping Reviews (PRISMA-ScR)

R: We have incorporated the acronym in the abstract in L36 and L37.

Comment #5:

L46-47 will be assessed for eligibility by three46 independent reviewers and discrepancies will be resolved by consensus. Seeking clarity: will consensus be from all 3 independent reviewers?

R: We have clarified the consensus between the reviewers. Now it reads like this in L43:

Assessment of eligibility after screening of titles, abstract and full text, and the resolution of discrepancies will be performed by three independent reviewers.

Comment #6:

L80: Sustainable Development Goals

Recommend: Inserting an acronym as well. Sustainable Development Goals (SDGs)

R: We have incorporated the acronym in L72.

Comment #7:

A definition of adolescents needs to be inserted into the introduction.

R: We have introduced a clear definition of adolescent. Now it reads like this in L80:

Another group at great risk are adolescents [11] (people from 10 to 19 years old), as they transition towards adulthood with significant physical, cognitive, emotional and social changes [12], including increased personal autonomy along with reduced parental supervision [13], and less normative or institutional protection

Comment #8:

L 127-8 There is currently no published synthesis127

on the effect of environmental factors on food insecurity among adolescents

Seeking clarity: Is this so globally? In Europe?

R: We have clarified the scope of available published synthesis. Now it reads like this in L143:

There is currently no published synthesis on the effect of environmental factors on food insecurity among adolescents from a global lens. 

Comment #8:

The objectives and the research questions should be separated.

R: We decided to be more precise and consistent in this point. Now, we use main objective and secondary objectives instead of research questions. Now it reads like this in L186:

The main objective is the understanding of the environmental factors that influence the experience of food insecurity in adolescents, following the structure of the “logic model of the problem” proposed in the IM methodology. Other secondary objectives include: 

- To identify the environmental factors (interpersonal, organizational, community and societal) that affect food insecurity in adolescents. 

- To analyze the mediating role of these factors on the food insecurity experienced by adolescents. 

- To describe the interventions addressing environmental factors of food insecurity, their gaps, limitations, and opportunities.

Comment #9:

Unsure if all study designs will be included.

R: We clarified the study designs that will be included. Now it reads like this in L228:

The review will include published studies (without restriction on study design) that provide information about the environmental factors of food insecurity on adolescents.

Comment #10:

L 216-27: The search strategy will be limited to literature published in English, Portuguese, or216 Spanish

Recommend: Insert a reason for only these three (3) languages being included.

R: We clarified the reason to only included studies publish in English, Portuguese or Spanish. Now it reads like this in L227:

(…) as our research team is fluent only in these languages.

Comment #11:

L229 -: reviewers will review titles and abstracts of studies for initial selection. Later, selected229 studies will go through a review of full text and data extraction by the same reviewers.230 The evaluators will regularly meet to discuss and resolve any disagreements. The results231 of the search will be detailed in the final report and presented in a flowchart (“PRISMA”).

Seeking clarity: Will all 3 reviewers resolve conflict/disagreements?

Reviewers, evaluators??? Try to be consistent

R: We are more consistent on the terms used to refer to the reviewers. Also, we clarified the role of all independent reviewers. Now it reads like this in L239:

Three independent reviewers will review titles and abstracts of studies for initial selection. Later, selected studies will go through a review of full text and data extraction by the same reviewers, that will also meet regularly to discuss and resolve any disagreements.

Comment #12:

This article should be sent to an editor/proofreader prior to it being resubmitted.

R: The manuscript has been revised by a native English speaker. 

Comment #13:

Additional information regarding screeners/funding etc. could be inserted.

R: We have added the funding information in a new section in L297:

The authors state that there is no funding source for the publication of this manuscript.

Reviewer #2:

Comment #1:

The authors have effectively positioned their subject (topic) as a relevant academic and policy endeavour by appropriately emphasizing its significance in the introduction and rationale of the study protocol.

They have substantiated the significance of comprehending environmental factors contributing to food insecurity in adolescents by noting that adolescence is a period of transition to adulthood characterized by reduced parental supervision and increased personal autonomy yet lacks comprehensive legislative or institutional safeguards.

Additionally, the authors have highlighted potential repercussions of food insecurity among adolescents. These consequences encompass physical health challenges such as wasting, stunting, nutritional deficiencies, anemia, and asthma, as well as impacts on academic performance and the risk of school dropout. Moreover, the protocol underscores the vulnerability of food-insecure adolescents to being compelled into child labor and early marriage, etc.

The reviewer recommends providing a more comprehensive explanation regarding the significance of this period, along with providing a clear insight into the potential ramifications of 'food stress' during this phase. This should be examined from both a public health perspective and an economic angle. The authors could elaborate on their argument along the following lines.

While early childhood is often recognized as the primary "window of opportunity" for learning and growth, adolescence introduces a distinct phase (referred to as the "second window of opportunity") that presents another critical timeframe for interventions. Throughout adolescence, individuals undergo rapid physical, cognitive, emotional, and social transformations. Consequently, any stress, including challenges related to food availability (food insecurity), and unhealthy eating habits experienced during this life stage can yield numerous health consequences (such as non-communicable diseases, overweight/obesity, etc.).

R: We thank the reviewer for her/his positive evaluation regarding the overall aim of our scoping review protocol. Also, thank you for her/his suggestion on providing a more comprehensive explanation about the significance of the adolescence and food insecurity. We also followed her/his proposal to examine the phenomenon from the public health and economic perspective. We have, accordingly, modified the following lines in the introduction to emphasize these points. 

First, we think we already state in the introduction the great threat that food insecurity in adolescence is for public health. We think so because we mention, not only the physical health consequences of food insecurity, but also the emotional, phycological, and social consequences. To make it more explicit we added the following in L105:

These consequences pose significant public health, social, economic, and human rights threats. 

Secondly, we developed further the idea of the economic threat adding information about the direct and indirect costs of food insecurity. It can be read like this in L106:

The costs include direct expenses for food assistance programs and clinical services for food-related conditions. Indirect costs stem from additional support and special education for children impacted physically or cognitively by food insecurity, affecting their academic performance and readiness for school [23].

Lastly, we incorporated your suggestion on using the term “second window of opportunity” while talking about the adolescence period. We also developed this idea in L83:

This "second window of opportunity" in development necessitates special attention, particularly in early adolescence (ages 10 to 14), to mitigate the negative consequences of food insecurity [14].

Comment #2:

The Method section appears to lack a well-structured organization in terms of its flow, structure, and content. The authors could have enhanced clarity by distinctly outlining each section to create a more formal protocol-like structure. Instead of writing as stage 1, stage 2, etc. the following sections could have been incorporated:

1. Objectives and Questions

2. Protocol Registration

3. Search Strategy

4. Inclusion and Exclusion Criteria

5. Quality Appraisal

6. Data Extraction

7. Data Analysis

R: Thank you for pointing this out. We agree with this comment, and we modified the sections accordingly. Now the “Methods and analysis” section is structured like this:

- Protocol design

- Conceptual model: “Logic model of the problem”

- Objectives

- Search strategy

- Study selection

- Data extraction and quality appraisal

- Data analysis

Additional clarifications:

In addition to the above comments, we modified the format of the abstract and removed the sections accordingly. Also, we clarified what are the environmental factors in the introduction rather the method section since we considered the readers would benefit from it being mentioned sooner in the manuscript. Now it reads like this in L111:

Addressing the environmental factors of food insecurity would require attention to their position at the interpersonal, organizational, community and societal level within socio-ecological models. The interpersonal level represents the individuals or groups –family and peers– closely connected to members of the priority population and with potential influence on their health-related behaviour. The organizational level refers to systems with specific objectives and multilevel decision-making processes, for example, schools and the healthcare system. The community level indicates the geographical area that comprises people and organizations; a social place shared by individuals that have a sense of living or working and common elements of values, culture, norms, language, and problems of health and quality of life. Finally, the society represents a larger system that controls several aspects of life and development of their constituent systems [24].

Under the same premise of thinking the readers would benefit from more clarity, we added a sentence before the section “Conceptual model: logic model of the problem” to be more precise on why we used the Intervention Mapping protocol. Now it read like this in L161:

This review is the first step to understand the phenomenon of adolescent food insecurity from a holistic perspective. It will take into account the environmental factors surrounding the adolescent, not only individual aspects.

Furthermore, we relocated the “Limitations” inside the “Discussion” section. It can be seen in page 12. Lastly, we have corrected the spelling and grammatical errors we detected while adding the reviewers’ comments, as well as making clearer some sentences.

---

## [Decision Letter · Decision Letter 1]

3 Nov 2023

Environmental factors of food insecurity in adolescents: a scoping review protocol

PONE-D-22-33353R1

Dear Dr. Capitán Moyano,

We’re pleased to inform you that your manuscript has been judged scientifically suitable for publication and will be formally accepted for publication once it meets all outstanding technical requirements.

Kind regards,

Sanjit Sarkar, PhD

Guest Editor

PLOS ONE

Additional Editor Comments (optional):

Reviewers' comments:

Reviewer's Responses to Questions

**Comments to the Author**

1. Does the manuscript provide a valid rationale for the proposed study, with clearly identified and justified research questions?

Reviewer #2: Partly

2. Is the protocol technically sound and planned in a manner that will lead to a meaningful outcome and allow testing the stated hypotheses?

Reviewer #2: Yes

3. Is the methodology feasible and described in sufficient detail to allow the work to be replicable?

Reviewer #2: Yes

4. Have the authors described where all data underlying the findings will be made available when the study is complete?

Reviewer #2: No

5. Is the manuscript presented in an intelligible fashion and written in standard English?

Reviewer #2: Yes

6. Review Comments to the Author

You may also provide optional suggestions and comments to authors that they might find helpful in planning their study.

Reviewer #2: 1. This scoping review protocol offers a well-founded rationale for the paper, along with clearly defined and justified objectives. The authors have effectively articulated the necessity for conducting a scoping review and have established precise objectives for their study.

2. This protocol provides a thorough account of the methods. While some aspects of the methodology and analysis may require refinement as the work progresses, the protocol is technically robust and aligns with the conceptual framework designed to achieve the stated objectives.

3. The methodology is outlined with sufficient detail, ensuring that the work can be replicated.

4. Given that English is not my native language, I am unable to provide a detailed linguistic evaluation. However, the manuscript appears to be presented in a comprehensible manner and is written in standard English, devoid of any noticeable typographical errors.

7. PLOS authors have the option to publish the peer review history of their article (what does this mean?). If published, this will include your full peer review and any attached files.

Reviewer #2: No

---

## [Editor Report · Acceptance letter]

13 Nov 2023

PONE-D-22-33353R1 

Environmental factors of food insecurity in adolescents:
a scoping review protocol. 

Dear Dr. Capitán-Moyano:

I'm pleased to inform you that your manuscript has been deemed suitable for publication in PLOS ONE. Congratulations! Your manuscript is now with our production department. 

Kind regards, 

on behalf of

Dr. Sanjit Sarkar 

Guest Editor

PLOS ONE